# Guaranteed Equal Opportunities? The Access to Nursing Training in Central Europe for People with a Turkish Migration Background

**DOI:** 10.3390/ijerph17124503

**Published:** 2020-06-23

**Authors:** Julia Keckeis, Margit Schäfer, Türkan Akkaya-Kalayci, Henriette Löffler-Stastka

**Affiliations:** Postgraduate University Program Transcultural Medicine, Medical University Vienna, 1090 Vienna, Austria; Julia.Keckeis@promente-v.at (J.K.); ulg-transkulturelleMed@meduniwien.ac.at (M.S.)

**Keywords:** migration and professional nursing, Turkish nurses, transcultural nursing, diversity in professional nursing

## Abstract

This paper examines the reason for the small percentage of professional nurses with a Turkish migration background and investigates possibilities to increase this low amount. Our society grows older, and the number of chronic diseases increases. Furthermore, nursing professionals tend to migrate, and the retirement of the baby boomer generation will also create a lack of professional nurses in Vorarlberg, Austria. People with a Turkish migration background, who are the second largest group without Austrian citizenship in Austria, could be an important resource for the upcoming lack of qualified nurses. The nursing profession could be a secure career opportunity for these people, and therefore it is of great importance to make access to professional nursing training easier for people with a Turkish migration background. This paper describes the effects of migration on society, institutions and individuals and gives an overview of concepts related to how to deal with this situation. This qualitative study investigates the access to nursing training for people with a Turkish migration background from three different points of view—those of experts, students and nurses with a Turkish migration background, and people with a Turkish migration background who have to pass a university entrance qualification—in the form of guided interviews. The results will illustrate structural and social barriers due to complex social dynamics and also highlight possibilities to reduce those barriers. Based on the results, prospects for professional nursing are deduced on the macro, meso and micro levels, which should generate an increasing number of nurses with a Turkish migration background.

## 1. Introduction

This study was carried out in Vorarlberg, the smallest federal state in Austria. A total of 66,230 out of 389,570 people living in Vorarlberg did not have Austrian citizenship in 2015; by the end of 2019, nearly 72,700 out of 397,852 people in Vorarlberg did not have an Austrian citizenship. The second largest group without Austrian citizenship was and still is people with Turkish nationality [1]. But this number does not include all the people living in Vorarlberg who have a Turkish background. This suggests that the number of people with a Turkish migration background is essentially higher, although explicit numbers were not found. Overall, the population in Austria has become older. In 2015, 18% of citizens were older than 65 years old; by the end of 2020, more than 20% of society in Vorarlberg will be over 65 years old [2]. Also, chronic diseases are increasing. In 2016, about 2.6 million people in Austria suffered from chronic diseases [3]. Another major point which underlines the upcoming necessity of professional nursing is the migration of nursing professionals. The shortage of applicants to nursing professions in Switzerland and Liechtenstein remove graduated nurses from Vorarlberg [4,5]. One of the biggest issues is that a huge number of nurses in Vorarlberg who belong to the baby boomer generation are going to retire in the next few years. The interplay of these issues mentioned above will lead to a deficit of 30,000 professional nurses in Austria by 2030 [6]. The well-known facts that a large number people with a Turkish migration background live in Vorarlberg and that the demand for qualified employees will grow exponentially raises the question why there are not more nursing professionals with a Turkish migration background. The background of this paper was not only an observation of the requisition of professional nurses, but it was also observed that there is a high heterogeneity in common nursing practice. The observation showed that people with a Turkish migration background often had insecurities, as they were confronted with misunderstandings and conflicts in their nursing team and in multi-disciplinary teams.

### 1.1. Professional Nursing in Austria

Professional nursing in Austria is strongly marked by the fundamental ideas of Christianity, for example charity as an unselfish altruism. During the nineteenth century, this basic concept was linked with conceptions of an ideal womanhood and the overall understanding of a woman’s role in society, therefore creating the perfect assistant to doctors [7]. The fundamentals for an independent profession was founded in 1997 with the Act on Nursing Care [8]. Academization started in 2016 with the amendment of this act [9]. This law describes graduate nursing care in the following way: “graduate nursing care is responsible for direct and indirect care for people of all ages, families and populations in mobile, outpatient, day care, and inpatient care, as well as all levels of care (primary care, outpatient specialist care and inpatient care). It is guided by ethical, legal, intercultural, psychosocial and systematic perspectives and principles” [9]. Furthermore, this act describes the different competences of professional nursing care, which are divided into three parts. The first section is one’s individual competence, which can be seen in evaluating how people establish or cope with risk factors, solve problems, deal with different requirements, and how they use resources. A second area is social communication and self-competence, which can be noticed in how people build up relationships, how they become structured, and how they come to a solution. The final area builds scientific expertise [10]. These competencies are quite demanding; nevertheless, young people tend to devalue the nursing profession. Stereotypical actions and tasks, such as basic care taking for other human beings and washing elderly people, encourage this opinion. Whereas older people mainly see the burden, younger people complain about the difficult framework and the low income of this profession [11].

### 1.2. Migration and the Effects on the Population

The reason for the high number of people with a Turkish migration background in Vorarlberg is the recruitment agreement of 1964 between Austria and Turkey [4]. The economy was the motor for this agreement, especially the textile, metal and construction industries. Turkish migrants worked in underpaid jobs with low social prestige. Outside the workplace, segregation was enforced between people from Turkey and natives from Vorarlberg [12]. It was not planned that these people settle down permanently, and in fact a lot of people returned to Turkey after a certain time. But after 1970, more and more people began to settle in Vorarlberg [13]. Today, there is still a high percentage of people with a Turkish migration background working in industry, but only 6.6% of these people have a university entrance qualification [14]. There are two major reasons for this: institutional organizations and individual factors, and their interaction [15].

### 1.3. Migration and Its Effects on Institutions

The diversity in society can also be observed within the hospital team and the patients. During everyday nursing care, language barriers are likely to occur. Also, misinterpretations of actions can result in conflicts or just in incorrect interpretations. This could lead to excessive demand, wrong treatments and even greater costs [16]. Therefore, different concepts were developed to deal with these problems in daily nursing care, for example, intercultural nursing, transcultural nursing, or transcultural competence, in which self-reflections, empathy for all individuals involved and the exchange of background knowledge are paramount [17]. The latest concepts are called cultural sensitivity and cultural sensibility nursing care, in which not only the acquisition of cultural knowledge is paramount. These concepts focus on empathy and empathy training, and not only to notice cultural features but also particularly to avoid putting them into stereotypical categories [18]. There are also concepts that were developed to handle and change the organizational tasks within care facilities. One is called intercultural opening, which strengthens the diversity in hospital or care-facility crews, as well as their training measures, in a culturally sensitive way [19]. A second concept is diversity management, which supports handling staff diversity. Diversity is unique and should be used purposefully and constructively to reach equality and equal treatment [20].

### 1.4. Migration and the Effects on Individuals

Migration often leads to a personal process of acceptance and identification. These effects can last over generations and sometimes entail personal conflicts. These people often struggle with different values, standards, and attitudes which have to be combined and processed, which in turn can result in physical and psychological illnesses [21]. Other aspects that have an effect on the health of people with migration backgrounds are social inequality, cultural differences, language barriers, lower chances in the labor market, as well their adverse housing situations and inconvenient overall living conditions [22]. The stereotype threat theory describes people who are worried about being confronted with negative stereotypes and their reaction to this threat, which often leads to retreat from public places. This also has effects on their academic position, careers, health, and integration [23]. In spite of these facts, migration also has positive effects. During the migration process, people often develop a bicultural identity. This means that people with bicultural identity relate to more than one social and cultural group. These people often have a larger degree of flexibility and adaptability and they are more open-minded to new situations, since these people are used to different ways of life [24].

### 1.5. Research Questions

From this background, the central question that motivates this paper is:

What possibilities exist to increase the number of professional nursing practitioners with a Turkish migration background?

Four additional questions were formulated to help obtain more differentiated scientific findings:(1)What are the experiences of experts in nursing training schools in Vorarlberg concerning people with a Turkish migration background during the admission procedure?(2)How do nurses and students with a Turkish migration background describe the admission procedure?(3)How is professional nursing seen by young people with a university entrance qualification and a Turkish migration background, and what are the attitudes of people in their social environment?(4)Which barriers can be derived and described on a micro-, meso-, and macro level according to the theory and the experiences of these three investigated groups?

### 1.6. Aim of the Study

The purpose of this research was to uncover barriers that occur in the entrance to nursing training schools for people with a Turkish migration background and to develop opportunities to overcome these barriers.

## 2. Methods and Material

### 2.1. Research Design

According to the Grounded Theory Technique in qualitative research design, a focus group of experts in the research field (2.1.1) and a literature search were used to define the interview/survey content, questions and the definition of the interviewees (2.1.2). For calculating the amount of interviews to be conducted, the saturation method (Hsieh and Shannon, 2005) was used.

#### 2.1.1. Research Field

The research field consisted of three different groups. The first group consisted of experts on nursing training (group E), which were determined by their profession and their extensive long-term experience in teaching and training nursing students. The second group (group 2) consisted of nurses with a Turkish migration background as well as nursing students with a Turkish migration background and the third group (group 3) consisted of possible candidates, i.e., people with a Turkish migration background who have a university entrance qualification.

#### 2.1.2. Access to the Research Field

Access to the research field was gained via the snowball effect while asking for potential interview partners from gatekeepers and informants. This snowball effect can distort data because of the aware and unaware preliminary selection of the interview partners. To avoid collecting distorted data, different criteria were defined and different gatekeepers were chosen. These were teachers for nursing training, headmasters of high schools, and the interview partners, as well as acquaintances. The interview partners were included according to a participatory research design.

### 2.2. Sample

According to the saturation method the sample size qA *n* = 10 and consisted of different characteristics, for example citizenship, age, place of birth, migration background, native language, religion, the highest graduation, and what kind of profession they pursue. There were two experts, two nurses with a Turkish migration background, three nursing students with a Turkish migration background, and three people with a Turkish migration background who have yet to pass a university entrance qualification.

### 2.3. Survey Method

The basic design of this paper was to obtain a snapshot of the apperception of all interviewed groups. The main issue was then analyzed from three different perspectives. Expert interviews were used to determine the process of the admissions procedure, to check how this procedure could be developed, and to reveal the relevant requirements for candidates. Experts’ positions towards the topic “access to nursing training schools for people with a Turkish migration background” were clarified [25]. For the other groups, a problem-focused interview was applied to obtain their opinions relating to this issue. In this type of interview, it is important to focus on narration. The guidelines are given by the interviewer to not lose the thread as well as to lead the interview in a direction that remains comparable [26,27].

### 2.4. Evaluation and Rating Method

The chosen final evaluation method was the deductive part according to the qualitative structuring content analysis by Mayring. To develop this rating and evaluation, first the data material was screened on content, aspects, and themes and then compressed into categories by two independent raters. An iterative process was conducted in order to diminish discrepancies and deepen understanding; this process was possible on the basis of a category system and encoding rules which were developed deductive and inductive. First, the analysis unit was defined. After that, categories and subcategories were labeled. These categories were subsumed, defined, they were made more differentiable by the help of encoding rules, and the categories were clarified by using examples. By means of these category systems, 10–50% of the data material was edited to prove this system. Afterwards, the category system was revised, and then the whole data material was screened. The places of discovery were extracted, paraphrased and generalized [28].

According to the local ethics committee of the country Vorarlberg, Austria, it was not necessary to obtain approval for this study from the ethics committee, as no patients were involved. But the informed consent of the interview partners had to be gathered according to data protection regulations. The consent of the interview partners was given by all participants. This was to stop the number of interviews at exactly the point when nothing new was gathered by the last interview, and therefore no added value can be expected by conducting further interviews (saturation method).

## 3. Results

Survey: According to the first research question (“What are the experiences of experts in nursing training schools in Vorarlberg/Austria concerning people with a Turkish migration background during the admission procedure?”) and the second question (“How do nurses and students with a Turkish migration background describe the admission procedure?”), the following information, descriptive results, and narrative content from the interviews could be retrieved in the survey.

### 3.1. Admission Procedure

The admission procedure has five steps: (1) a digital application has to be made which is followed by (2) an interview with the headmaster of nursing training schools. After that, (3) an internship has to be attended. The fourth step (4) is an admission test which can have various forms. This test is followed by (5) an interview in front of an admission committee. These five steps are compulsory for all nursing trainees. Nevertheless, the interviews and the admission test are not standardized.

The experts had rarely experienced people with a Turkish migration background in past admission procedures. Therefore, they did not see any difficulties for people with a Turkish migration background, if the above-mentioned steps are fulfilled.

Nevertheless, individual students and nurses with a Turkish migration background report that they have been discriminated and confronted with prejudice during the admission procedure. The whole group that was interviewed reported differences in the admission procedures between the three nursing schools in Vorarlberg. Differences can be found in organization and assessment and, depending on the nursing school, whether the educational background was focused on or not.

### 3.2. Individual Respondent’s Source of Information and Support for Choosing a Nursing Education

The respondents obtained their information from direct experiences with nurses and from their own social environment. They were also supported by their social environment in the form of information as well as financial and mental support from the families. The internet, as well as information events, were also mentioned to provide them with further information and were helpful to confirm an opinion.

Evaluation and rating (first two questions): Concerning the deductive/inductive rating process, due to content analysis, the following categories could be condensed concerning the first two research questions.

### 3.3. Interest and Motivation

The more fine-grained content analysis of the data (Figure 1) reveals that interest is awoken best when people are in direct contact with nursing professionals. These experiences are combined with positive memories and are therefore relevant. Interestingly, advertising had no major impact on people. The main motivation to embark on a professional nursing education is an overall social attitude with the idea of helping others or spending time with individuals as well as gaining autonomy. Other motivation factors are career possibilities, the prospect of a secure job, diverse opportunities, thankful patience, and the opportunity to work in a team (see also Table 1).

The third research question (“How is professional nursing seen by young people with a university entrance qualification and a Turkish migration background, and what is the attitude of the social environment?”) and the fourth question (“Which barriers can be derived and described on a micro-, meso-, and macro level according to the theory and to the experiences made by the three investigated groups?”) showed the following attitudes in the survey.

### 3.4. Barriers

The interviews show that the nursing profession has a low local value and therefore people do not consider the nursing profession as an opportunity (see also Figure 2). People often have an incorrect image of this job. They tend to see the profession in a very stereotypical way, with images in their mind like washing elderly people and assisting doctors that are also typically gender-related female stereotypes. These old-fashioned gender roles still exist. For people with a Turkish migration background, the experience of discrimination and prejudice can lead to an overall struggle to perform well in such a public profession. The interviewed people also mentioned a sense of shame as a barrier to engage and act in this profession.

The following quotes illustrate such attitudes that could form certain conflicts: *“they can dress as they want in class, but in practice they should wear the prescribed uniforms.” (E2, line 513–515)*; or at least show ambivalence: *“And some, they really push into our culture. … and others, there I just have the feeling that they use our system, but they do nothing. … So, I think it will be a social task to differentiate. Really to see who wants.” (E2, line 842–851)*.

Within the content analytic process of the material gathered within all four research questions, the overall research aim (“What possibilities exist to increase the number of professional nursing practitioners with a Turkish migration background?”) could be answered as follows.

### 3.5. Possibilities to Overcome These Barriers

The results of the present study demonstrate that the respondents see the lack of information concerning the nursing profession as the main barrier. Information should be given on various channels. Most respondents (Figure 3) agreed that specific advertising with visual or linguistic indicators are a good possibility to show their openness for people with a Turkish migration background. To quote one respondent, *“When you see … would see … that you are sought after” (B1, line 1091)*. The most noteworthy result of all the interviews was that society should be informed more precisely and there should be a special focus on informing people with a Turkish migration background. Nursing schools should cooperate with gatekeepers in order to access communities. The number of role models in care facilities and nursing training schools who have a Turkish migration background should be increased. Again, a quote from a respondent illustrates the latter: *“I mean, we just live together and there are also many Turkish people who have to go to the hospital … and when more help from the Turkish people is offered, e.g., if you do not speak German. … It is of course easier if you have more Turkish staff or one other nation, does not necessarily have to be Turkish.” (B2, line 514–519).*

## 4. Discussion

### 4.1. Awareness of the Profession and Attitudes

The social environment of the nurses, students and people with university entrance qualifications with a Turkish migration background see the nursing profession as necessary, exhausting but also as accountable. The nursing profession is often seen as a primitive care-giving job. Often it is combined with images like washing people which can be disgusting for many people. It is also seen as unskilled labor and as a typical female profession in which the potential earnings are low. These statements are similar to the remarks of Görres [11] in respect of the images of professional nursing. These examples prefigure the low status of this profession. However, nursing training is believed to be very difficult. The reputation of the admission processes is a tough one with high requirements. In turn, students at nursing schools and nurses with a Turkish migration background also see a change in the image of this profession in their environment. This could have been most likely the result of an interaction between the interview partners and their environment.

### 4.2. Barriers

The reputation of this profession seems to be one of the barriers encouraging the low number of people with a Turkish migration background in the nursing profession. In the interviews, the barriers were mentioned on structural and social levels. On the structural level, it was emphasized that the priority of the German language, educational background, economic conditions, and the lack of role models, as well as the general handling of diversity in the schools, are the major barriers. There were no references of concepts or strategies for diversity management or intercultural opening in relation to patience and the hospital staff, nor in schools for the students and the faculty. On the social level the sociocultural background (education, values and norms, religion, sex), the awareness of the nursing profession and their status, the knowledge and information as well as one’s personal and influenced experiences play the biggest role. One of the most important points which was highlighted during the interviews was the lack of information. This could be an explanation for the low status of the nursing profession and the still dominant image of a typical female vocation. In addition, prejudice and discrimination experiences are huge barriers. The interviewed experts do not seem to be aware of these points, which also show how significant this issue is to them. Similarly, Burtscher-Mathis [15] describes the problems of students with a Turkish migration background in the complete education system in Austria. He also states the interrelation of institutional impact and individual influential factors as a reason for the educational disadvantage of students with a Turkish migration background in Austria. The results of this study also show that there are social, institutional and individual influences which can become barriers. These influences underlie complex social dynamics which cannot be seen separately. The main obstacle on the micro level is the absence of interest and motivation. This absence of interest could be a result of the lack of information. Other barriers are the status of the profession, all the negative experiences people with a Turkish migration background underwent in society, as well as all their socio-economic conditions. On the meso level, they operate with public relation in the form of advertising. This advertising does not seem to reach people with a Turkish migration background. The admissions procedures of all the nursing training schools in Vorarlberg attempt to evaluate every candidate in the same way under the cloak of equality and neutrality. This could lead to a distorted view of reality because unequal conditions are not considered, which in return reproduces inequality. But the biggest problem on the meso level is the absence of a discussion about diversity and its consequences in all nursing training schools, and their spread advertisement as well as the admissions procedures. Our education system is implicated in some shortfalls which lead to one of the biggest barriers on the macro level. Society, which includes politics and all other public institutions, tend to occur as a barrier because of their acquaintance of how to deal with diversity in general. This can also be seen in today’s political and media situation.

### 4.3. Possibilities to Increase the Number of Nurses with a Turkish Migration Background

The data reveals that the sensitization of all stakeholders is the main way to create awareness in this issue in nursing training schools, hospitals and other care facilities. Awareness is the key to find new possibilities and remove these barriers. Nursing training schools as well as all other care facilities should deal with the subject of diversity. Students, patients, and also staff members should be part of that discussion. Therefore, Spiel [29] claims to aspire to adequate heterogeneity by teachers and to bundle knowledge about diversity in order to arrive a resource-oriented way of thinking for the whole Austrian education system. Furthermore, it is important that the institutions reflect their own values and attitudes to make room for change. The results of the present study demonstrate that this issue receives low attention in various institutions. There are different theories to handle this topic which were designed for institutions, for example, diversity management or intercultural opening. An important consideration for specific public relations is to illustrate the openness and interest of people with a Turkish migration background in nursing professions. This could be done by visual indicators in advertising and information about competencies and opportunities in this profession through different channels. While developing these public relation methods, the social network of the target group should be considered as an important guide, and therefore they should be kept clearly in mind. While cooperating with Turkish associations and gatekeepers, the nursing profession should be presented as a good option to awaken interest. Specific actions to support the applicants could be a mentoring system as well as the organization of preparation courses.

### 4.4. Courses of Action

The data seem to suggest that it is important to integrate the social environment as a source of information on the micro level, because this is one of the most essential information resources for people. This is also reflected in career choice research, which shows the influence of the social network [30]. The entrance to the micro level could happen with specific advertising on different channels. Gatekeepers and target actions could be used to awaken interest in this profession as well as to support people as they apply for nursing training schools. Target actions could be a mentor system or preparation courses. At the meso level, sensitization to this issue in all different institutions and the reflection of their expectations, attitudes, prejudices, and values is the basis upon which to achieve a rearrangement. Therefore, concepts like intercultural opening or diversity management could be useful. Furthermore, it is necessary to change the image of the nursing profession in society. This includes how professionals present themselves and their daily work in public. Speaking generally, it would be useful to have a better connection between all kinds of institutions, for example care facilities, schools, professional association, and labor unions. Another essential point is to offer try-out-days and internships to make sure that the interested participants are well informed. That means that internships should be experience-oriented and not only operation-oriented. On the macro level, politics and the media should discuss in an objective way topics such as migration and religion. The overall task for our education system concerning language and compensating factors to minimize unequal conditions should be evaluated. Schools should also focus on different mechanisms of role assignment and sex division in future occupational career chances.

### 4.5. Experiences during Admission Procedure

Experts in nursing training schools report that they have hardly any experiences with people with a Turkish migration background as participants in the admission procedure in the past. They perceive an underrepresentation of people with a Turkish migration background in the nursing profession and in nursing training schools. Furthermore, they declare that people with a Turkish migration background in nursing training school are not more prominent than other people. Out of the group which consisted of nurses and students with a Turkish migration background, individuals reported discrimination experiences. As there are many actors involved in an admissions process, personal opinions vary. All actors have their own ideas about people with a Turkish migration background, so it would be very important that these ideas are reflected upon from time to time to avoid negative stereotypes and prejudice.

### 4.6. Concluding Limitations, Strength and Outlook

This paper is based on a qualitative research which aims to find new ideas in a hypotheses-generating way. The goal was not to prove hypotheses, and because of the small number of interview partners this paper is not representative. Although this paper is not representative, it should give an idea of the institutional and social barriers which occur in the admissions procedure to nursing training schools for people with a Turkish migration background and opportunities to conquer them. A number of restrictions had to be performed during this study. This, on the other hand, allows for further research. Investigations in how to handle discrimination against students, nurses and patients with migration backgrounds in hospitals, schools and care facility could be performed. Furthermore, how the care-taking quality could be harmed by negative stereotypes and prejudice on the part of the nursing professionals and their consequences could be invested. Further opportunities should be sourced and focused to reduce frustration, to avoid resignation, and to remove prejudice within the nursing profession in relation to people with migration backgrounds. In particular, nursing training schools should prepare their students for diverse sociocultural belongings in daily nursing practice.

## 5. Conclusions

All points mentioned in this paper lead to the fact that the recruitment of nurses with a Turkish migration background has positive effects on society because they could be a resource for the upcoming shortage of professional nurses. Care facilities and hospitals benefit from nurses with migration background not only as employees but also because they have a mediation function. Besides, the quality of care could be better granted, which helps to reduce costs and allows institutions to fulfill their function as role models in society. Moreover, people with migration backgrounds receive another opportunity for a secure job; the quality of nursing for people with migration background would increase; and the entrance for people with migration background to professional nursing would become easier. Therefore, the nursing profession could contribute the implementation of Vorarlberg’s integration model.

## Figures and Tables

**Figure 1 ijerph-17-04503-f001:**
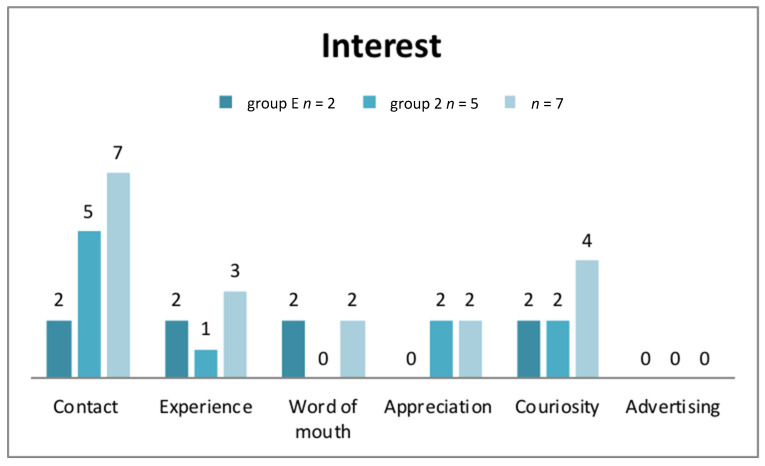
Results on how to awaken interest on this profession.

**Figure 2 ijerph-17-04503-f002:**
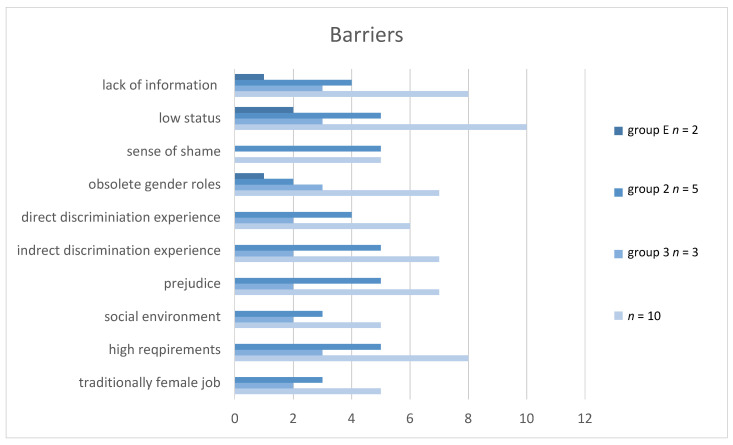
Results on the barriers during the access to nursing training school.

**Figure 3 ijerph-17-04503-f003:**
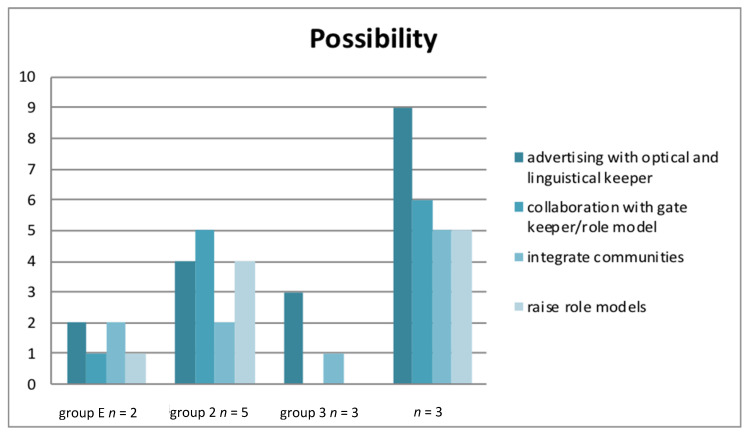
Results on how to overcome the barriers.

**Table 1 ijerph-17-04503-t001:** Results on the motivation to work as a professional nurse.

Extrinsic Motivation	Group E	Group 2	Group 3	*n* = 10
	*n* = 2	*n* = 5	*n* = 3	
Career possibilities	2	1	1	4
View of a secure job	1	3	0	4
Potential earnings	2	2	2	6
Thankfulness/appreciation	0	2	2	4
**Intrinsic Motivation**				
Social attitude	2	4	2	8
Ambition	1	4	0	5
Financial independence	0	3	1	4

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
