# Peer review of "Guaranteed Equal Opportunities? The Access to Nursing Training in Central Europe for People with a Turkish Migration Background"

_ijerph, 2020, doi:10.3390/ijerph17124503_

Round 1

Reviewer 1 Report

Dear Authors,

Thank you for sending through your revised manuscript.  You have addressed the reviewers comments well.  I have just made a couple of further minor points for clarification.

Thanks

Line 10 - Change chronical diseases to "chronic diseases"

Line 30 - change 66.230 to 66,230 or 66 230 - this change from . to , should be made throughout the paper

Line 193 - could the authors please state the "local ethics committee's" name and it would be useful to view the documentation/email correspondence

Line 205 - could the authors please clarify the five steps for the admission procedure.  The current description is confusing.

Line 239 - table 1 - could the authors please clarify "Possibilities" as a motivation?

Line 360 - change sources to resources

Line 362 - Please clarify "network (Eckert, 2008)proof"

Author Response

Open Review

English language and style

( ) Extensive editing of English language and style required 
( ) Moderate English changes required 
(x) English language and style are fine/minor spell check required 
( ) I don't feel qualified to judge about the English language and style 

Yes

Can be improved

Must be improved

Not applicable

Does the introduction provide sufficient background and include all relevant references?

(x)

( )

( )

( )

Is the research design appropriate?

(x)

( )

( )

( )

Are the methods adequately described?

( )

(x)

( )

( )

Are the results clearly presented?

(x)

( )

( )

( )

Are the conclusions supported by the results?

(x)

( )

( )

( )

Comments and Suggestions for Authors

Dear Authors,

Thank you for sending through your revised manuscript.  You have addressed the reviewers comments well.  I have just made a couple of further minor points for clarification.

Thanks

Line 10 - Change chronical diseases to "chronic diseases"

Line 30 - change 66.230 to 66,230 or 66 230 - this change from . to , should be made throughout the paper

Line 193 - could the authors please state the "local ethics committee's" name and it would be useful to view the documentation/email correspondence

Line 205 - could the authors please clarify the five steps for the admission procedure.  The current description is confusing.

Line 239 - table 1 - could the authors please clarify "Possibilities" as a motivation?

Line 360 - change sources to resources

Line 362 - Please clarify "network (Eckert, 2008)proof"

  • Thank you for the helpful hints, we changed according to the suggestions and marked it yellow.

Submission Date

01 June 2020

Date of this review

15 Jun 2020 22:38:54

Reviewer 2 Report

To all that contributed to this paper: this paper is now interesting and well designed. It can contribute to this kind of research in this field. 

All the best

Author Response

Thank you!!

This manuscript is a resubmission of an earlier submission. The following is a list of the peer review reports and author responses from that submission.

Round 1

Reviewer 1 Report

Thanks for your submission.  It is interesting, but I would like to see the following revisions made and the article resubmitted.  There are numerous spelling and grammatical errors contained within the article.

Line 12 - Please state the country that Vorarlberg is in.

I would like to read the reasons why the focus is on Turkish nurses stated in the abstract, although you have stated the reasons in the Introduction.

I wonder if there is more recent demographic data of the region, compared to 2015

Line 76 - could you clarify the term "a lot of"?

Line 96 - space is required after categories

Lines 118-128 - I would suggest that the research questions are included in a table or on their own line so that its clear what the questions are.

Line 136 - clarification about 'experts' - who are they and how have they been determined?

Line 140 - This section covers recruitment - More specif details are required.

Could the authors please confirm the ethical clearances that where obtained for this study?

Line 173 - clarification about who are the headmasters is needed

The methods section requires more details including the interview questions and process of analysis.  This requires revision.

The results (section 3) are poorly presented.  This section requires revision and inclusion of data so the reader can understand what was collected

Quotes from the interviews would be useful to illustrate the results.

It is hard to interpret the the discussion without clear results being presented.

Line 278 - I am unsure the meaning of the following sentence - Data reveal that the sensitization of all actors is the main access to create awareness in nursing training schools, hospitals and the other care facilities - please clarify

Line 281 - stuff should be staff

Line 281, 282 and 283 should be rewritten - the sentence structure and meaning is confusing.

Line 383 - proof should be prove

I would suggest that the limitation of this study is included.

The discussion should relate to the results presented and existing literature.  I would suggest that this section is revised.

Author Response

Open Review

English language and style

( ) Extensive editing of English language and style required 
(x) Moderate English changes required 
( ) English language and style are fine/minor spell check required 
( ) I don't feel qualified to judge about the English language and style 

Yes

Can be improved

Must be improved

Not applicable

Does the introduction provide sufficient background and include all relevant references?

(x)

( )

( )

( )

Is the research design appropriate?

(x)

( )

( )

( )

Are the methods adequately described?

( )

( )

(x)

( )

Are the results clearly presented?

( )

( )

(x)

( )

Are the conclusions supported by the results?

( )

( )

(x)

( )

Comments and Suggestions for Authors

Thanks for your submission.  It is interesting, but I would like to see the following revisions made and the article resubmitted.  There are numerous spelling  and grammatical errors contained within the article.

  • Many thanks for the helpful suggestions for improvement of our manuscript. We corrected style and grammar throughout the whole manuscript.

Line 12 - Please state the country that Vorarlberg is in.

  • We now stated the country additionally, thank you for the hint. Please see lines 12-13

I would like to read the reasons why the focus is on Turkish nurses stated in the abstract, although you have stated the reasons in the Introduction.

  • Many thanks for this suggestion, we added the reasons, that they belong to the second largest group without Austrian citizenship already in the abstract (line 13) and added this information in the first paragraph of the introduction.

I wonder if there is more recent demographic data of the region, compared to 2015.

  • We added recent data in line 31-32 and the relevant reference (2020)

Line 76 - could you clarify the term "a lot of"?

  • The term is clarified and specified explaining the background (line 78-80).

Line 96 - space is required after categories.

  • The space is inserted now. Please see line 100.

Lines 118-128 - I would suggest that the research questions are included in a table or on their own line so that its clear what the questions are.

  • The research questions are stated now clearly at the end of the introduction to give an overview. Please see lines 123-134

Line 136 - clarification about 'experts' - who are they and how have they been determined?

  • Experts are defined more precisely in the methods section. Please see lines 149-151

Line 140 - This section covers recruitment - More specify details are required.

  • Also the recruitment of the interviewees is added in differentiating the whole research design. Please see lines 144-168

Could the authors please confirm the ethical clearances that where obtained for this study?

  • We added a paragraph concerning the informed consent. Please see lines 193-197

Line 173 - clarification about who are the headmasters is needed.

  • Headmasters of nursing training school and high-schools are added. Please see line 206

The methods section requires more details including the interview questions and process of analysis.  This requires revision.

  • The methods section is described in more detail now, describing the specific steps. Please see lines 144-197

The results (section 3) are poorly presented.  This section requires revision and inclusion of data so the reader can understand what was collected

à The results section is restructured now, refers to the research questions and is grouped into the specific steps of evaluation procedure. Please see lines 199-217, 225-227, 232, 241-245, 262-271 and 282-286

Quotes from the interviews would be useful to illustrate the results.

  • Thank you for this excellent suggestion, we added some quotes in the results section to illustrate barriers and possibilities. Please see lines 262-267 and 282-286

It is hard to interpret the the discussion without clear results being presented.

à The discussion is now grouped according to the refined and restructured results.

Line 278 - I am unsure the meaning of the following sentence - Data reveal that the sensitization of all actors is the main access to create awareness in nursing training schools, hospitals and the other care facilities - please clarify

à Thank you for the hint, it is clarified and changed.

Line 281 - stuff should be staff à corrected.

Line 281, 282 and 283 should be rewritten - the sentence structure and meaning is confusing. It is rewritten, please see 344-346

Line 383 - proof should be prove. à corrected.

I would suggest that the limitation of this study is included.

à Thank you, a limitation and outlook to further investigation possibilities is included now.

The discussion should relate to the results presented and existing literature.  I would suggest that this section is revised.

à The discussion is revised and refers now to the literature. Many thanks for the suggestions.

 Submission Date

12 February 2020

Date of this review

26 Feb 2020 18:24:02

Reviewer 2 Report

Even if the argument is a bit interesting it is an "old" issue and there have been already several papers on the matter and at European level several projects and research. In particular, the old Erasmus and the new Erasmus plus program has one of its main domain on the integration of immigrates. Also several "European discussion tables" have produced recommendations and guidelines for this issue, even if not specific.

Author Response

Open Review

English language and style

( ) Extensive editing of English language and style required 
( ) Moderate English changes required 
(x) English language and style are fine/minor spell check required 
( ) I don't feel qualified to judge about the English language and style 

Yes

Can be improved

Must be improved

Not applicable

Does the introduction provide sufficient background and include all relevant references?

(x)

( )

( )

( )

Is the research design appropriate?

( )

( )

(x)

( )

Are the methods adequately described?

( )

(x)

( )

( )

Are the results clearly presented?

( )

(x)

( )

( )

Are the conclusions supported by the results?

( )

(x)

( )

( )

Comments and Suggestions for Authors

Even if the argument is a bit interesting it is an "old" issue and there have been already several papers on the matter and at European level several projects and research. In particular, the old Erasmus and the new Erasmus plus program has one of its main domain on the integration of immigrates. Also several "European discussion tables" have produced recommendations and guidelines for this issue, even if not specific.

  • Thank you very much for pointing at this topic. Although there are programs on the Erasmus level, the Turkish students have to have a Visum as their country is not a Schengen country. These Visa last only for 90 days which is too short for practical training nursing students would need.

Submission Date

12 February 2020

Date of this review

17 Feb 2020 15:53:17
